

# Chemical characterization of laboratory-generated tar ball particles

Ádám Tóth[1], András Hoffer[2], Mihály Pósfai[1], Tibor Ajtai[3], Zoltán Kónya[4,5], Marianne Blazsó[6], Zsuzsanna Czégény[6], Gyula Kiss[2], Zoltán Bozóki[3], András Gelencsér[1,2]

[1]Department of Earth and Environmental Sciences, University of Pannonia, Veszprém, P.O. Box 158, H-8201, Hungary
[2]MTA-PE Air Chemistry Research Group, Veszprém, P.O. Box 158, H-8201, Hungary
[3]MTA-SZTE Research Group on Photoacoustic Spectroscopy, Szeged, Dóm tér 9, H-6720, Hungary
[4]Department of Applied and Environmental Chemistry, University of Szeged, Szeged, Rerrich Béla tér 1, H-6720, Hungary
[5]MTA-SZTE Reaction Kinetics and Surface Chemistry Research Group, Szeged, Rerrich Béla tér 1, H-6720, Hungary
[6]MTA-TTK Institute of Materials and Environmental Chemistry, Budapest, P.O. Box 286, H-1519, Hungary

*Correspondence to:* András Gelencsér (gelencs@almos.uni-pannon.hu)

**Abstract.** The chemical properties of tar ball (TB) particles generated from dry distillate (wood tars) of three different wood species in the laboratory were investigated by analytical techniques that had never been used before, for their characterization. The elemental composition of TB particles from three tree species were very similar to one another and to those characteristic for atmospheric tar balls (TBs) collected from savanna fire during the SAFARI 2000 sampling campaign. The O/C and H/C molar ratios of the generated TBs were at the upper limit characteristic for soot particles. The FT-IR spectra of the generated TBs were very similar to one another as well and also showed some similarity with those of atmospheric humic-like substances (HULIS). The FT-IR measurements indicated that laboratory-generated TBs have a higher proportion of aromatic structure than HULIS and the oxygen atoms of TBs are mainly found in hydroxyl and keto functional groups. Whereas the starting materials of the TBs (wood tars) were Raman inactive in the range of 1000–1800 $cm^{-1}$, the Raman spectra of TBs were dominated by two pronounced bands with intensity maxima near 1580 (G band) and 1350 $cm^{-1}$ (D band), indicating the presence of $sp^2$-hybridised carbon structures and disorder in them, respectively. In the Py-GC-MS chromatograms of the laboratory-generated TBs mostly aromatic compounds (aromatic hydrocarbons, oxygenated aromatics and heterocyclic aromatics) were identified in accordance with the results of Raman and FT-IR spectroscopy. According to OC/EC analysis using EUSAAR_2 long thermal protocol, 22% of the total carbon content of laboratory-generated TBs was identified as elemental carbon (EC), contrary to expectations based on the current understanding that negligible if any EC is present in this sub-fraction of the brown carbon family. Our results suggest that spherical atmospheric TBs with high C/O molar ratios are closer to BC in many of their properties than to weakly absorbing HULIS.

## 1 Introduction

Atmospheric tar balls (TBs) comprise a unique class of carbonaceous aerosol particles emitted during biomass burning (Pósfai et al., 2003; 2004; Adachi and Buseck, 2011). TBs are understood to be part of the family of atmospheric brown carbon (BrC) (Hand et al., 2005; Andreae and Gelencsér, 2006), as they absorb light in the visible range of the solar spectrum yet are distinctly different from BC in microstructure, morphology and in other properties (as summarized by Petzold et al. (2013)). Since these particles are fairly abundant in biomass burning plumes and are able to absorb solar radiation quite efficiently in the visible (Hand et al., 2005; Alexander et al., 2008) and up to the near-IR range (Hoffer et al., 2017), TBs may have a considerable effect on the Earth radiation budget (Chung et al., 2012). TBs can be unambiguously identified by electron microscopy as perfectly spherical amorphous particles externally mixed in relatively fresh biomass burning plumes (Pósfai et al. 2003 and 2004, Adachi and Buseck, 2011). Unlike soot particles, TBs do not form chain-like aggregates of 20–50 nm spherules and there are no turbostratic / concentrically wrapped graphitic layers in their microstructure. Their sizes range from 30 to 500 nm in geometric





diameter (Pósfai et al., 2004). Furthermore, TBs are refractory as they can withstand the high-energy electron beam of the TEM in vacuum (Pósfai et al., 2004, Hand et al, 2005). In addition, elemental composition (C/O molar ratio) is also an important characteristic of TBs. Typical C/O molar ratios of atmospheric TBs are about 9–10, as determined by transmission electron microscopy with energy-dispersive X-ray spectroscopy (TEM-EDS). It should be noted that in some studies the term 'tar ball' is

used for combustion particles that are non-spherical and have lower C/O molar ratio (Chakrabarty et al., 2016). However, in this paper we use the term exclusively to refer to combustion particles that share all the key characteristics that were described above. Albeit TBs are abundant in biomass burning plumes globally, very little is known about their chemical composition mainly because in biomass smoke TBs coexist with various other particle types (e.g., organic particles with inorganic inclusions, soot) from which they cannot be separated physically. Thus, the chemical properties of TBs can only be studied by single particle

analytical techniques such as TEM-EDS (Li et al., 2003; Pósfai et al., 2003; 2004; Hand et al., 2005; Niemi et al., 2006; Adachi and Buseck, 2011; Chakrabarty et al. 2016; Adachi et al., 2017), TEM with electron energy-loss spectroscopy (TEM-EELS) (Hand et al., 2005, Adachi and Buseck, 2011), scanning electron microscopy with energy-dispersive spectroscopy (SEM-EDS) (Li et al., 2003; Hand et al., 2005; Cong et al., 2009; 2010; China et al., 2013; Chakrabarty et al., 2006; 2010), environmental transmission electron microscopy (ETEM) (Semeniuk et al., 2006), environmental scanning electron microscopy (ESEM) (Hand

et al., 2005), and near-edge X-ray absorption fine-structure spectroscopy (NEXAFS) using a synchrotron source (Tivanski et al., 2007).

In our previous works (Tóth et al., 2014; Hoffer et al., 2016) an experimental setup has been developed for the generation of TB particles in the laboratory without the concurrent emission of other combustion products. The morphological and structural characteristics and the elemental composition of the TB particles generated by this experimental system were highly similar to

those of atmospheric TBs. In the present study the chemical properties of the TBs produced in the laboratory were investigated by several analytical techniques which have never been applied for the characterization of TBs. The analytical methods deployed were direct elemental analysis (CHNSO), OC/EC thermal-optical analysis (TOA), pyrolysis-gas chromatography-mass spectrometry (Py-GC-MS), Fourier transform-infrared spectroscopy (FT-IR) and Raman spectroscopy. The results of the analyses were directly compared to those obtained by other studies using the same techniques for atmospheric humic-like

substances (HULIS), or humic acid and soot (BC) with a view to locate TBs in the light-absorbing carbon continuum.

## 2 Experimental section

The generation of TB particles in the laboratory were carried out as described in Hoffer et al. (2016). Briefly, TBs were produced from the liquid tarry condensates obtained by dry distillation of wood chops of Norway spruce (*Picea abies*), European turkey oak (*Quercus cerris*) and black locust (*Robinia pseudoacacia*), separately. The concentrated aqueous phase of the tarry

condensates (wood tars) was nebulised to produce tar droplets which were first exposed to a 'thermal shock' by passing them through a heated (at 650 °C) quartz tube, then cooled and dried with dry filtered air. The optical and the morphological properties of the particles generated from Norway spruce and black locust were described by Hoffer et al. (2016). The shapes of the TB particles (investigated by TEM) generated from European turkey oak were mostly distorted spheres. In this case the majority of the particles were likely deliquescent upon collision with the collection surface. The morphologies of these particles were very

similar to those of the freshly formed atmospheric TBs (Adachi and Buseck, 2011), and the size distribution and optical properties were similar to those published in Hoffer et al. (2016). The generated particles were collected on different sampling substrates: on TEM grids (lacey Formvar/carbon TEM copper grid of 200 mesh, Ted Pella Inc., USA), on pre-baked double (front and backup) quartz filters (QMA, ø 47 mm, Whatman) and on pre-cleaned aluminium foils using a Berner cascade





impactor (Wang and John, 1988). In the case of the impactor samples we used the samples collected on stage 2 (aerodynamic diameter between 125 and 250 nm), representing about half (~44%) of the mass of the generated particles. The morphologies and the elemental compositions (C, O, N, S) of individual TB particles from different wood species were studied in bright-field TEM images obtained using a Philips CM20 TEM operated at 200 kV accelerating voltage. The electron microscope had an attached

ultra-thin-window Bruker Quantax X-ray detector that allowed the energy-dispersive X-ray analysis (EDS) of individual particles. The relative concentrations of C and O were determined using sensitivity ratios (k-factors) derived from EDS spectra acquired from a standard (a stoichiometric dolomite ($CaMg[CO_3]_2$) sample). Nitrogen was not determined quantitatively.

The elemental (C, H, N, S, O) compositions of the aqueous phases of the concentrated wood tar samples (the starting material for TB generation) and those of the laboratory-generated TBs were determined using an EuroVector EA3000 CHNS/O elemental

analyzer. The carrier gas was helium (He) (purity: 4.6; Messer) with a flow rate of 110 L min$^{-1}$, the temperature of the reactor tube and that of the GC oven were 980 °C and 70 °C, respectively. The instrument was equipped with a TCD detector. The measurements of CHNS and oxygen content of samples were carried out separately from quartz filters and all analyses were performed in duplicate. The portions of filters with area of 1 cm$^2$ were packed in double tin (ø 5 × 9 mm) capsules in the case of the CHNS analysis while in the case of the oxygen analysis the samples were wrapped in double silver capsules (ø 5 × 9 mm).

The data were corrected for blanks taken from backup quartz filters of the same size in double tin/silver capsules. The calibration for these elements was carried out using reference standard materials (acetanilide, SOIL#5, SOIL NCS–2) from EuroVector, Italy.

The OC/EC thermal-optical analysis of generated TBs was performed by a Model-4 Semi-Continuous OC-EC Field Analyser (Sunset Laboratory Inc., USA). The aerosol samples on quartz filters (ø 13.06 mm) were analysed following the EUSAAR_2

long protocol. The data were corrected for blanks taken from backup quartz filters of same size.

The TB samples collected on quartz filters were investigated using Py-GC-MS. The analyses were performed with a Pyroprobe 2000 pyrolyzer (CDS Analytical) interfaced directly to a gas chromatograph-mass spectrometer (Agilent 6890A/5973). The portions of sample filters with areas of 0.5 cm$^2$ were heated from 250 °C to 600 °C at a heating rate of 1 °C ms$^{-1}$, and held for 20 s in the pyrolyzer. High-purity He (purity: 5.0; Linde) as carrier gas was used at a controlled flow rate of 20 mL min$^{-1}$ to flush

the pyrolysis products into a DB-1701capillary column (30 m × 0.25 mm ID, 0.25 μm film thickness, Agilent). The GC injector was set in splitless mode with an inlet temperature of 250 °C. The temperature of the column was kept at 40 °C for 2 min, then increased at a heating rate of 10 °C min$^{-1}$ to 280 °C and held there for 5 min. Temperatures of the GC-MS interface and the detector were 280 °C and 230 °C, respectively. The mass spectrometer was operated at 70 eV with a mass detection in the m/z range of 15–350.

The characteristic functional groups of wood tar and laboratory-generated TB samples collected on aluminium foils were examined using specular reflection FT-IR technique. The spectra of the samples were recorded on a Bruker Vertex 70 FT-IR spectrometer coupled with a Hyperion 2000 IR microscope with 15× (NA = 0.4) specular reflection objective. Spectra were recorded over the range of wave number 4000–400 cm$^{-1}$ at room temperature using 128 scans at 2 cm$^{-1}$ resolution.

The wood tar and laboratory-generated TB samples collected on aluminium foils were also investigated by Raman spectrometry.

Raman spectra were recorded with a Thermo Scientific DXR Raman microscope at excitation wavelength ($\lambda_0$) of 532 nm, applying max. 10 mW laser power, with the laser beam focused using a 50× objective lens, resulting in a spot size of ~1 μm. Typically, 20 scans were recorded and averaged with 4 cm$^{-1}$ resolution in the 200–1800 cm$^{-1}$ range.



## 3 Results

### 3.1 Elemental composition of laboratory-generated TBs

According to the CHNSO analysis the mean C, H, N and O contents of TB samples (n = 3) on quartz filter were 82 % (RSD: 0.5 %); 4 % (RSD: 6.7 %); 3 % (RSD: 39 %); and 11 % (RSD: 9.2 %) by mass, respectively. Sulphur was below the detection limit.

Table 1 summarizes the average O/C and H/C molar ratios of wood tars and TBs produced from the three wood types, as determined by CHNSO elemental analysis. For comparison, the O/C molar ratios obtained for individual particles by TEM measurements are also given. The O/C molar ratios of the laboratory-generated TB particles obtained from three different tree species were very similar to each other and to the values obtained from TEM-EDS analyses. The H/C molar ratio was relatively low (between 0.51 and 0.58), indicating that the laboratory-generated TBs consist mostly of unsaturated, aromatic and
oxygenated organic compounds. It should be noted that wood tars (starting material for TB generation) exhibited significantly higher O/C and H/C molar ratios (0.182 and 1.215, respectively), which strongly suggests that the 'thermal shock' employed during TB generation (as described in Hoffer et al. (2016)) has markedly increased the degree of aromatisation (Francioso et al., 2011).

In order to compare the elemental composition of laboratory-generated TBs with those of soot, HULIS and atmospheric TBs, a
van Krevelen diagram is plotted (Fig. 1). For the TBs measured in earlier studies using TEM-EDS (Pósfai et al., 2004), SEM-EDS (Chakrabarty et al., 2010; China et al., 2013) and NEXAFS (Tivanski et al., 2007), only the available O/C molar ratios are presented in the diagram. It can be clearly seen that the average O/C molar ratio of our laboratory-generated TB particles is very similar to that of atmospheric TBs examined by Pósfai et al. (2004), whereas it is lower than those obtained by some other authors (Tivanski et al., 2007; Chakrabarty et al., 2010; China et al., 2013). This difference in compositions may result from
differences in the formation temperatures and/or atmospheric processing of the TBs. China et al. (2013) collected slightly aged (1–2 h) particles from the smoldering phase of the Las Conchas fire in northern New Mexico, USA. Chakrabarty et al. (2010) investigated particles from the smoldering combustion of dry duffs, whereas Tivanski et al. (2007) observed aged TBs during episodes characterized by high particle light-scattering coefficients. In contrast to these authors, Pósfai et al. (2003 and 2004) measured TB particles from both flaming and smoldering savanna fires, although the authors mention that the distinction
between different burning stages was often not straightforward, since flaming and smoldering stages of the burn could be present simultaneously in adjacent areas. The uncertain identification of burning stages notwithstanding, these observations suggest that several types of TBs may exist with different O/C molar ratios, depending on the formation temperature and the temperature and duration of the heat shock that the particles are exposed to, and/or on the degree of atmospheric processing.

However, when comparing the elemental compositions of our laboratory-generated TBs to that of HULIS, both the O/C (0.094–
0.109) and the H/C (0.511–0.584) molar ratios of our TBs were substantially lower than those reported for HULIS samples (O/C: 0.455–0.563; H/C: 1.431–1.537) (Krivácsy et al., 2001; Kiss et al., 2002; and Salma et al., 2007). The O/C molar ratio of other atmospheric TBs varies widely and in some cases compares better with the O/C molar ratio of HULIS. The O/C and H/C molar ratios of TBs identified by Pósfai et al. (2004) from savanna fires, as well as our laboratory-generated TBs (O/C: 0.094–0.109; H/C: 0.511–0.584) are close to the upper limit of those characteristic for soot (O/C: ~0.12; H/C: ~0.67).

As expected from the above results, the mean carbon to mass conversion factor of our TBs (1.21; RSD: 0.5%) is between that of HULIS (1.81–1.93; Krivácsy et al., 2001; Kiss et al., 2002; Salma et al., 2007) and soot samples (1.04–1.15; Akhter et al., 1985; Clague et al., 1999).



### 3.2 Characterization by FT-IR spectroscopy

The FT-IR spectra of wood tar and laboratory-generated TB samples are characterised by broad and overlapping bands (Figure 2). The figure also shows that the FT-IR spectra of TBs produced from the three different wood species are much more similar to one another than the spectra from different wood tars. Large differences in the IR spectra of wood tars can be observed

particularly in the fingerprint region (1400–500 cm$^{-1}$); in contrast, being exposed to a heat shock, the laboratory-generated TBs from different sources became chemically similar to one another.

The FT-IR spectra of wood tars and TBs show a very broad band between 3600 and 3000 cm$^{-1}$ (assigned to OH-stretching of phenol and/or hydroxyl groups) and a smaller band in the region between 3000 and 2780 cm$^{-1}$, attributed to asymmetric and symmetric C–H stretching of methyl and methylene aliphatic groups (Coates, 2000; Graber and Rudich, 2006; Yang et al.,

2007). In the spectra of TBs the sp$^2$-aromatic C–H stretching at 3060 cm$^{-1}$ (Coates, 2000; Cain et al., 2010; Santamaria et al., 2006) is more pronounced than in the spectra of wood tars, indicating the increased aromaticity. The spectra of wood tars and TBs are dominated by two strong bands at ~1700 cm$^{-1}$ and at ~1605 cm$^{-1}$, assigned to C=O stretching and C=C stretching of aromatic rings (with overlapping C=O stretching), respectively (Coates, 2000; Graber and Rudich, 2006; Santamaria et al., 2006; Cain et al., 2010). The ratio of these two bands is the opposite in the two sample types, the intensity of the aromatic C=C

stretching increases relative to the C=O stretching in TBs.

The region between 1450 and 1380 cm$^{-1}$ can be assigned to aliphatic or aromatic methyl and methylene bending (Craddock et al., 2015; Coates, 2000; Santamaria et al., 2006; Cain et al., 2010).

The aromatic C–C and C–H plane deformation bands in the region between 1300–1000 cm$^{-1}$ overlap with the band of the C–O single bond. The broad band at 1220 cm$^{-1}$ probably belongs to the C–O stretching of phenolic hydroxyl groups in FT-IR spectra

of wood tar and TB samples (Coates, 2000; Yang et al., 2007), whereas the peaks at ~920; ~1040, ~1110 and ~1321 cm$^{-1}$ correspond to the C–H bending of carbohydrate; to C–O stretch in the C–OH in carbohydrate structure; to stretching of the C–O of the C–O–C linkage; and O–H bending of C–OH group, respectively (Santamaría et al., 2006; Yang et al., 2007; Cain et al., 2010; Carletti, et al., 2010; Anjos et al., 2015).

By comparing the IR spectra of TBs with those of HULIS it can be concluded that they show large-scale similarity, since the

characteristic bands, the aliphatic and aromatic C–H, aromatic C=C, hydroxyl and keto groups (Krivácsy et al., 2001; Kiss et al., 2002; Duarte et al., 2005; Graber and Rudich, 2006) are present, but the intensity ratios of the C=O (~1700 cm$^{-1}$) and the C=C (~1605 cm$^{-1}$) bands are the opposite. Thus, laboratory-generated TBs have a higher proportion of aromatic structure than HULIS and the composition of HULIS is more similar to the wood tars in this respect. On the other hand, HULIS spectra contain characteristic features (a broad band at 3400–2400 cm$^{-1}$ and a band of C=O at ~1700 cm$^{-1}$) suggesting the presence of carboxyl

groups, whereas these bands cannot be found in the TB spectra. Another difference between the spectra of TBs and HULIS is that the HULIS spectra contain a very broad band (assigned to O–H stretching in carboxyl group), which occurs at 3400 to 2400 cm$^{-1}$ and often overlaps with C–H stretching. Since this characteristic broad band is missing, and the band of C=O (~1700 cm$^{-1}$) appear at lower frequencies than 1720 cm$^{-1}$ in the spectra of both wood tar and laboratory-generated TB samples, it is assumed that they do not contain carboxyl groups.

The FT-IR spectra of TBs also differ from those of soot: the band representative of the acetylenic group at 3300 cm$^{-1}$ is absent in the spectra of TBs and the spectra of soot do not contain the bands of OH-stretching (Cain et al., 2010; Santamaria et al., 2006; 2010).





### 3.3 Raman spectroscopy

Raman spectroscopy was used to characterise the short-range order in the molecular structure of laboratory-generated TBs. All three types of wood tar were Raman inactive in the range of 1000–1800 cm$^{-1}$, whereas the Raman spectra of laboratory-generated TBs were dominated by two pronounced bands with intensity maxima near 1580 and 1350 cm$^{-1}$. This double peak was

deconvoluted by the five band fitting procedure first proposed by Sadezky et al. (2005), but it was found that instead of using four Lorentzian (G, $D_1$, $D_2$, $D_4$) and one Gaussian ($D_3$) peaks, the best fit was obtained with five Voigt functions, similarly to Catelani et al., (2014). The peak fitting of Raman spectra (in the range between 1000 and 1800 cm$^{-1}$) were executed after multi-point baseline correction using by the GRAMS/AI (Version: 7.02) software. The Raman spectra and examples for the peak deconvolution (in the range between 1000 and 1800 cm$^{-1}$) of the laboratory-generated TBs are shown in Figure 3. The Raman

spectra of the TBs generated from Turkey oak were not evaluated, since the peak fitting was uncertain due to a reduced signal-to-noise ratio. The presence of the G band in the Raman spectra of laboratory-generated TBs indicates that TBs contain an aromatic layer built up from sp$^2$-hybridised carbon atoms, whereas the existence of the D bands points to the presence of poorly organised carbonaceous materials. It is important to note that the macromolecular humic acid (purified, Carl Roth GmbH, Karlsruhe; Germany) investigated by Ivleva et al. (2007) proved to be also Raman active in this range, with Raman spectra very similar to

those of our laboratory-generated TBs.

### 3.4 Py-GC-MS measurements

With Py-GC-MS of the laboratory-generated TB samples approximately 40 compounds were identified which are listed in Table 2. The pyrolysis products were identified by comparison of their mass spectra with the standard mass spectra in the NIST 02 Library (NIST/EPA/NIH Mass Spectral Library, 2002).

In the pyrograms of the laboratory-generated TB samples mainly aromatic compounds have been identified in accordance with the results of Raman and FT-IR spectroscopy, and indirectly with the results of the elemental analysis. Aromatic hydrocarbons (benzene, alkyl-, alkenyl-substituted benzenes and smaller (2–3 aromatic rings) polycyclic aromatic hydrocarbons (PAHs), oxygenated aromatics (phenol, alkyl-substituted phenols) and heterocyclic aromatics (phthalic anhydride, furan, benzofuran, dibenzofuran and their derivatives) were identified. Many of the above-mentioned components have been identified using the

same analytical technique from humic acid (extracted by sodium hydroxide solution and precipitated with hydrochloric acid from urban aerosol) and different (hexane, gasoline, diesel and wood) soot samples (see Table 2) in previous studies (Subbalakshmi et al., 2000, Song and Peng, 2010). Comparing the chromatograms of humic acid, TBs and soot, significant differences occur in the quality (different numbers of aromatic rings) and quantity (number of compounds) of PAH components. The pyrograms of humic acid contain only few small (2-ring) PAHs, e.g., naphthalene and its derivatives (methyl- and dimethyl-naphthalene),

whereas in the pyrograms of the laboratory-generated TBs larger (3-ring) PAHs can be also found. On the other hand, in the pyrograms of soot samples (Song and Peng, 2010) both smaller (2–3-ring) and larger (4–5-ring) PAHs were also found, but the latter compounds were not detected in the pyrograms of laboratory-generated TBs.

### 3.5 OC/EC thermal-optical analysis

Since TBs belong to the BrC fraction of carbonaceous aerosol (Hoffer et al., 2016), their EC content is expected to be very small

or even negligible. Since the results of Raman spectroscopy indicated some structural similarities with atmospheric soot, we determined the apparent EC content of laboratory-generated TBs by standard OC/EC analysis. The results of OC/EC thermal-optical analysis of TBs which were produced from three different wood species are given in Table 3.



The EC/TC ratio for laboratory-generated TBs varied from 0.17 to 0.32, (on average: 0.22; RSD: 39.1%) which is far from being negligible, contrary to expectations. It is also important to note that there is an uncertainty in the position of the split point in the OC/EC measurement. In the case of the TBs the criteria that OC should be non-absorbing is not met; thus, the absorption of BrC lowers the baseline of the transmittance. Consequently, the split point is set earlier and a larger EC signal is measured (Chen et
al., 2015). The thermogram of TBs produced from Norway spruce shows that the detector signal does not return to the baseline after the 4$^{th}$ OC peak, and neither after the pyrolytic carbon peak, and thus a notable fraction of pyrolytic carbon (PC) is identified as EC (see Figure 4).

## 4 Conclusions

We have studied the chemical properties of laboratory-generated TBs using analytical techniques that had never been used for
the characterization of atmospheric TBs, and compared the results with those obtained for soot, HULIS and humic acid. The elemental compositions of the TB particles generated from different wood species were very similar to one another and to those characteristic for atmospheric TBs formed in savanna fires. In the van Krevelen diagram (plot of O/C vs. H/C molar ratio) soot and HULIS are confined to rather narrow regimes and are distinctly different from TBs which have highly variable O/C molar ratios, depending on the conditions prevalent during their formation and/or subsequent atmospheric processing. The O/C and
H/C molar ratios of the laboratory-generated TBs (and the atmospheric TBs identified from savanna fires) are much lower than those of HULIS and closer to the upper bound characteristic for soot. The FT-IR spectra of the different laboratory-generated TBs were also similar to one another and also to some extent to the spectra of HULIS. However, the oxygen atoms in laboratory-generated TBs are found to be largely present in components of keto and hydroxyl functional groups, with very few if any in carboxyl groups occurring in TBs, unlike in HULIS. While the starting material of TBs (wood tars) were Raman inactive in the
range of 1000–1800 cm$^{-1}$, the appearance of G (characteristic for ordered sp$^2$-hybridised carbon structures) and D bands (characteristic for structural disorder in sp$^2$-hybridised carbon systems) in the Raman spectra of laboratory-generated TBs indicated the presence of some short-range order in molecular structures. In the Py-GC-MS pyrograms of different laboratory-generated TBs mostly aromatic compounds were identified. While pyrograms of laboratory-generated TBs did not indicate the presence of larger condensed aromatic structures (with 4–5 aromatic rings) which are characteristic for soot, they showed several
naphthalene derivatives and 3-ring condensed aromatics which are absent among the pyrolysis products of HULIS. Using the EUSAAR_2 long thermal protocol, on average 22% of the total carbon of the laboratory-generated TBs was found to be elemental carbon (EC), contrary to the expected negligible if any EC in a sub-fraction of the brown carbon family. Our results have demonstrated chemical differences between wood tars and TBs, confirming the formation mechanism proposed by Tóth et al. (2014), and have helped to position various types of TBs along the organic-to-graphitic carbon continuum of combustion
aerosols. In this regard, the combination of all analytical results presents an array of supporting chemical evidence that spherical atmospheric TBs with C/O molar ratio around 10 are closer to BC in many of their properties than to weakly absorbing HULIS. In harmony with the findings of several independent studies on the optical properties of TBs, the present results imply that TBs are indeed quite strongly light-absorbing aerosol particles and likely play an important role in the global radiation budget.

## Competing interests

The authors declare that they have no conflict of interest.



**Acknowledgments**

This paper was supported by the János Bolyai Research Scholarship of the Hungarian Academy of Sciences. The work was supported by the project GINOP-2.3.2-15-2016-00055. The project is realized with the support of the European Union, with the co-funding of the European Social Fund. This work was also supported by the project GINOP-2.3.2-15-2016-00036 and EFOP-
3.6.1-16-2016-00014. The authors would like to thank K. L. Juhász and V. Havasi for their assistance with this project.

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



**Table 1. Oxygen to carbon (O/C) and hydrogen to carbon (H/C) molar ratios of laboratory-generated TBs and wood tar samples measured by TEM/EDS (from analysis of 12 particles from each sample) and by CHNSO elemental analyser. *Data from Hoffer et al., (2017).**

|  | O/C molar ratio (by TEM-EDS) | O/C molar ratio (by CHNSO) | H/C molar ratio (by CHNSO) |
|---|---|---|---|
| TB - Black locust | 0.110 (12%)[*] | 0.094 | 0.584 |
| TB - Norway spruce | 0.108 (7.4%)[*] | 0.109 | 0.511 |
| TB - Turkey oak | 0.111 (9.1%) | 0.094 | 0.543 |
| TB average | 0.110 (9.5%) | 0.099 (8.9%) | 0.546 (6.7%) |
| Wood tar samples average | no data | 0.182 (5.9%) | 1.215 (1.4%) |





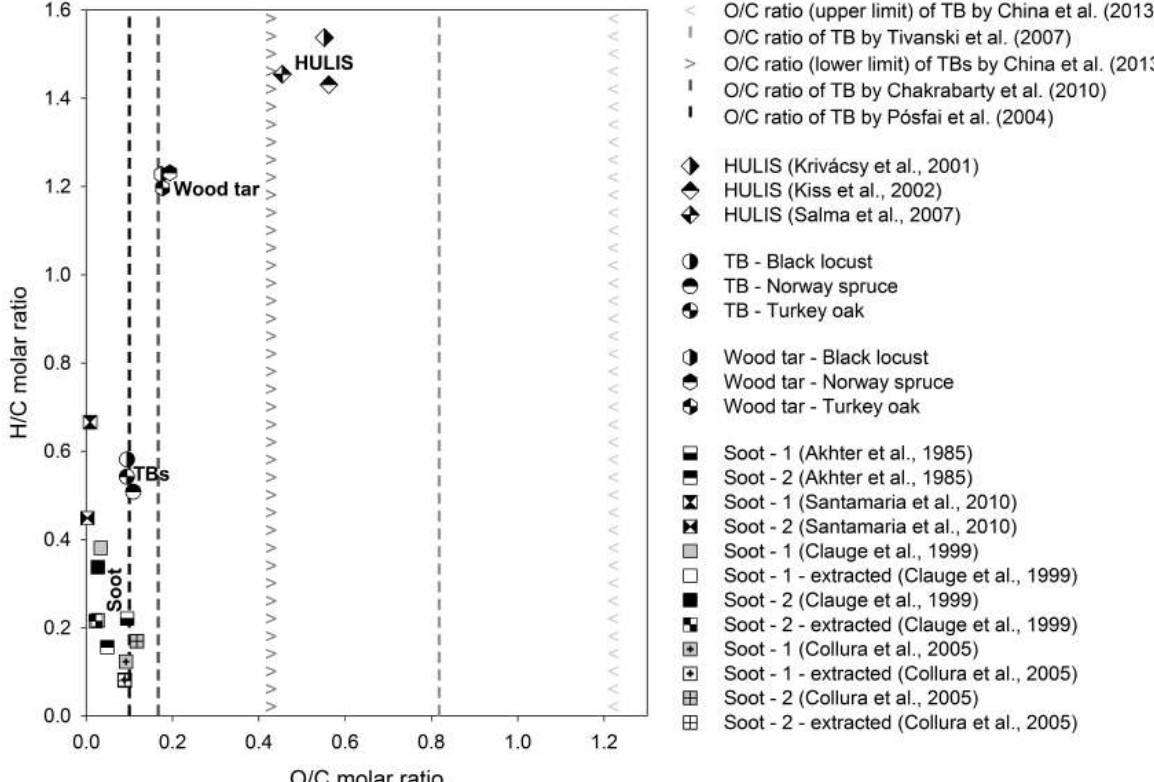

**Figure 1.** Van Krevelen diagram of different soot (Akhter et al., 1985; Clague et al., 1999, Collura et al., 2005; Santamaria et al., 2010), TB, wood tar and HULIS (Krivácsy et al., 2001; Kiss et al., 2002; and Salma et al., 2007) samples.





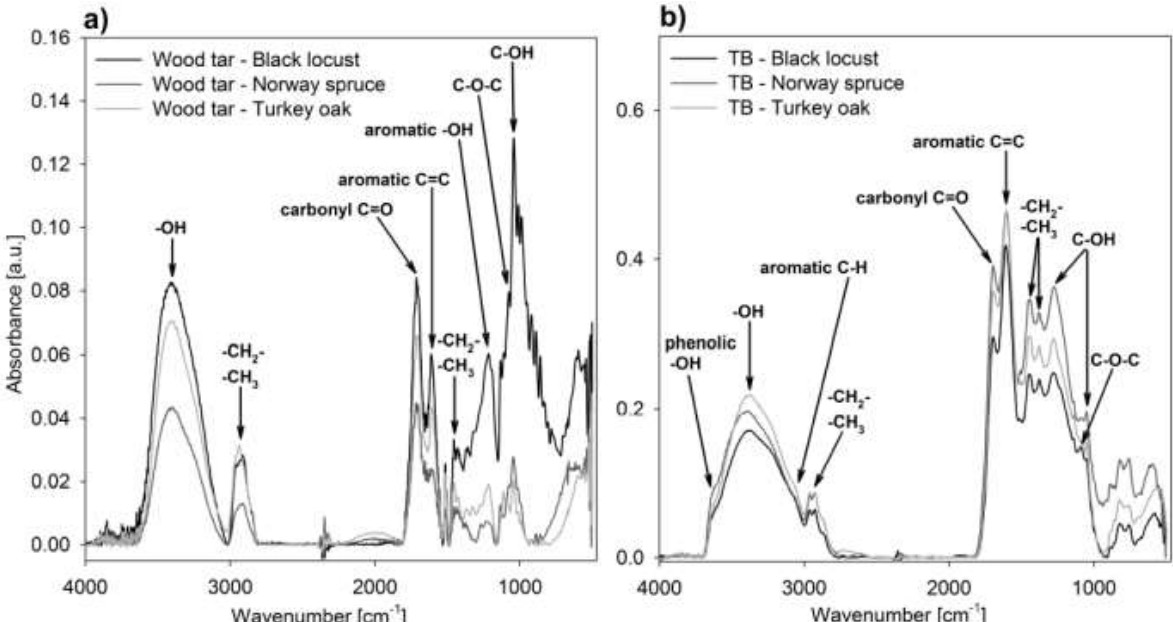

Figure 2. FT-IR spectra of (a) wood tars and (b) laboratory-generated TBs produced from different wood species.





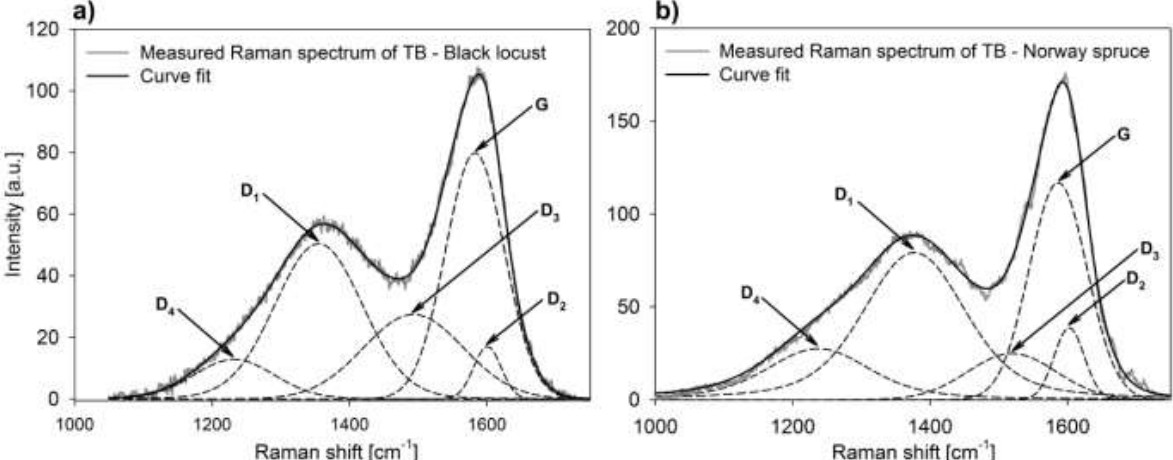

**Figure 3. Curve fit with five bands for the first-order Raman spectra (excitation wavelength: $\lambda_0 = 532$ nm) of tar ball (TB) particles, produced from (a) black locust and (b) Norway spruce, using as proposed by Catelani et al. (2014) for carbonaceous materials.**



**Table 2. Identified components from Py-GC-MS chromatogram of TBs produced from different wood species, compared with the identified pyrolytic products of hexane, gasoline, diesel, wood soot (Song and Peng, 2010), and humic acid extracted from particulate matter (Subbalakshmi et al., 2000).**

| Name of pyrolytic compounds | Examined samples | | | Reference samples | | | | |
|---|---|---|---|---|---|---|---|---|
| | TB Black locust | TB Norway spruce | TB-Turkey oak | Soot Hexane | Soot Gasoline | Soot Diesel | Soot Wood | Humic acid |
| AROMATIC HYDROCARBONS | | | | | | | | |
| ➢ 1-RING AROMATIC HYDROCARBONS | | | | | | | | |
| Benzene | × | × | × | | | | | × |
| o/m/p-Dimethylbenzene | × | ×* | × | × | × | × | × | × |
| α-Methylstyrene | × | × | ×* | | | | | × |
| Styrene | × | ×* | ×* | × | × | × | × | × |
| Toluene | × | × | × | × | × | × | × | |
| ➢ 2-RING AROMATIC HYDROCARBONS | | | | | | | | |
| Biphenyl | × | × | × | × | × | × | × | × |
| 1,6-Dimethylnaphthalene | × | × | × | | | | | × |
| 2,3-Dimethylnaphthalene | × | × | × | | | | | × |
| 2,7-Dimethylnaphthalene | × | × | × | | | | | × |
| Indene | × | × | × | × | × | | | |
| 2-Methylindene | × | × | × | | | | | |
| 1-Methylnaphtalene | × | × | × | × | × | | × | |
| 2-Methylnaphtalene | × | × | × | × | × | | × | |
| 3-Methyl-1H-indene | × | × | × | | | | | |
| Naphtalene | × | × | × | × | × | × | × | |
| ➢ 3-RING AROMATIC HYDROCARBONS | | | | | | | | |
| Acenaphthylene | × | × | × | × | | | | |
| Anthracene | | | | × | × | | × | |
| Fluorene | × | × | × | × | × | | × | |
| 1-Methyl-9H-fluorene | | | × | | | | | |
| 4-Methyl-9H-fluorene | | | × | | | | | |
| Phenanthrene | | | | × | × | × | × | |
| ➢ 4-RING AROMATICS HYDROCARBONS | | | | | | | | |
| Fluoranthene | | | | × | × | | × | |
| Pyrene | | | | × | × | × | × | |
| ➢ 5-RING AROMATIC HYDROCARBONS | | | | | | | | |
| Benzo[mno]fluoranthene | | | | × | × | | × | |
| OXYGENATED AROMATICS | | | | | | | | |
| ➢ 1-RING OXYGENATED AROMATICS | | | | | | | | |
| Acetophenone | × | × | × | | | | | |
| Benzaldehyde | × | ×* | ×* | | | | | × |
| 2,4-Dihydroxy-3,6-dimethylbenzaldehyde | | × | × | | | | | |
| 2,3-Dimethylphenol | × | × | × | | × | | | |
| 2,4-Dimethylphenol | × | × | × | | × | | | |
| 2,5-Dimethylphenol | × | × | × | | × | | | |





| | | | | | | | | |
|---|---|---|---|---|---|---|---|---|
| 2,6-Dimethylphenol | | × | × | | × | | | |
| 3,4-Dimethylphenol | × | | | | × | | | |
| 2,6-Dimethoxyphenol | | | | | | | | × |
| 2-Ethylphenol | | × | × | | | | | × |
| 2-Methoxy-4-methylphenol | | | | | | | | × |
| 2-Methoxyphenol | | | | | | | | × |
| 2-Methylphenol | × | × | × | | × | | | × |
| 4-Methylphenol | × | × | × | | × | | | × |
| Phenol | × | × | × | × | × | × | × | × |
| ➤ 2-RING OXYGENATED AROMATICS | | | | | | | | |
| 2,3-Dihydro-1H-Inden-1-one | ×* | × | ×* | | | | | |
| Phthalic acid anhydride | × | × | × | | | | | × |
| OXYGEN-CONTAINING HETEROCYCLIC AROMATICS | | | | | | | | |
| ➤ 1-RING OXYGEN-CONTAINING HETEROCYCLIC AROMATICS | | | | | | | | |
| 2,4-Dimethylfuran | ×* | | ×* | | | | | |
| 3-Furancarboxaldehyde | | | × | | | | | × |
| 5-Methyl-2-furaldehyde | ×* | × | ×* | | | | | × |
| ➤ 2-RING OXYGEN-CONTAINING HETEROCYCLIC AROMATICS | | | | | | | | |
| 2-Methylbenzofuran | × | × | × | | | | | |
| 7-Methylbenzofuran | × | × | × | | | | | |
| Benzofuran | × | × | × | | × | × | × | |
| ➤ 3-RING OXYGEN-CONTAINING HETEROCYCLIC AROMATICS | | | | | | | | |
| Dibenzofuran | × | × | × | | | | × | |

*: The concentration of the given component is the same as on the back up filter in this sample.



**Table 3. Organic carbon (OC), elemental carbon (EC), and total carbon (TC) content of tar balls (TBs) on quartz filter (spot ø 13.06 mm) obtained by the EUSAAR_2 long protocol.**

|  | OC [$\mu g\ cm^{-2}$] | EC [$\mu g\ cm^{-2}$] | TC [$\mu g\ cm^{-2}$] | EC/TC |
|---|---|---|---|---|
| TB - Black locust | 9.0 | 4.2 | 13.2 | 0.32 |
| TB - Norway spruce | 14.1 | 2.9 | 17.1 | 0.17 |
| TB - Turkey oak | 14.2 | 2.9 | 17.1 | 0.17 |
| TB average |  |  |  | 0.22 (RSD: 39 %) |



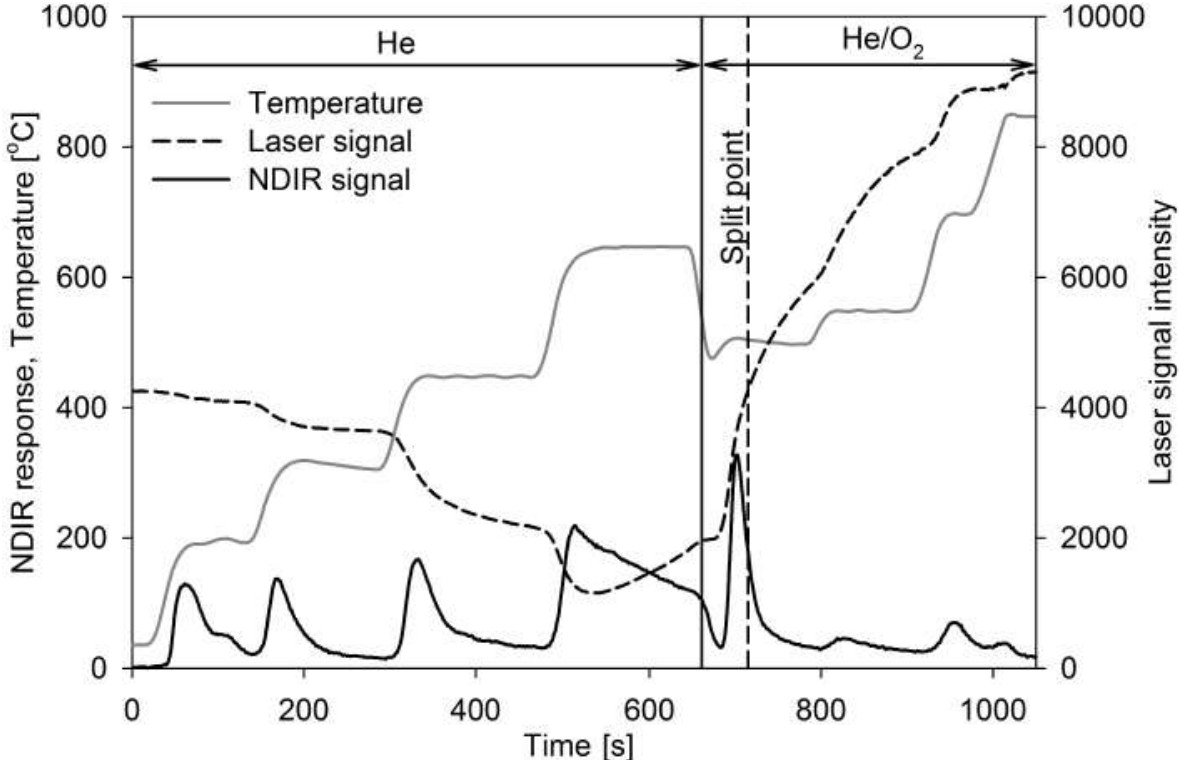

**Figure 4. Thermogram from analysis of TBs produced from Norway spruce, measured by thermal-optical analysis (TOA) obtained by the EUSAAR_2 long protocol.**