# Peer review of "Chemical characterization of laboratory-generated tar ball particles"

_Atmospheric Chemistry and Physics, 2018_

## Referee Comment (RC1) · Anonymous Referee #1 · 30 Mar 2018

General comments:

This study, "Chemical characterization of laboratory-generated tar ball particles", by Tóth et al., generated tar balls and characterized their chemical properties using several unique instruments. They compared tar balls with other similar carbonaceous particles (BC and HULIS) and discussed how they are different or similar. Tar ball particles are important and abundant in the atmosphere but are not well known. Thus, the characterization of tar balls is significant in this field. The manuscript is clearly written with enough data. I feel more discussion is useful regarding the similarity of the atmospheric and laboratory-generated tar balls, in addition to their morphology and O/C molar ratio. Overall, I think this manuscript includes important results on the atmospheric science.

Specific comments:

[Figure]

1. Page 2 line 18-20: The morphological and structural characteristics and the elemental composition of the TB particles generated by this experimental system were highly similar to those of atmospheric TBs.

Reviewer comment: Please discuss more about how they are similar to each other. Their similarity is a critical point to use the laboratory-generated tar ball data to the atmospheric tar balls. Although some such discussion may have been done in their previous studies, I think some discussion is useful here. Also, the tar ball formation process in this laboratory experiment may be different from that observed in other studies from wild fire (e.g., Posfai et al., 2004; Adachi and Buseck, 2010). They found more tar balls in aged smoke than young smoke. The laboratory-generated tar balls are emitted as primary particles and can be detected directly from the fresh smoke, suggesting that no tar ball number change as aged(?). More explanation will be useful here how they are similar to or different from ambient tar balls regarding their formation process.

2. Page 2 line 32-34: The shapes of the TB particles (investigated by TEM) generated from European turkey oak were mostly distorted spheres.

Reviewer comment: Tar balls are defined as "TBs can be unambiguously identified by electron microscopy as perfectly spherical amorphous particles externally mixed in relatively fresh biomass burning plumes" in Page 1 line 35-36 or "in this paper we use the term exclusively to refer to combustion particles that share all the key characteristics that were described above" in page 2 line 5-6. However, "distorted sphere" contradicts "perfectly spherical amorphous particles." Clear definition of tar ball will be needed here, if they include those are not perfectly spherical. TEM images of European turkey oak tar ball may be useful as well as those from other sources.

3. Page 3 line 11.

Reviewer comment: Please spell out "TCD."

4. Page 4 Line 10-13. It should be noted that wood tars (starting material for TB generation) exhibited significantly higher O/C and H/C molar ratios (0.182 and 1.215, respectively), which strongly suggests that the 'thermal shock' employed during TB generation (as described in Hoffer et al. (2016)) has markedly increased the degree of aromatisation (Francioso et al., 2011).

Reviewer comment: Is it possible that water (H2O) is included in the wood tar to increase these molar ratios?

5. Page 4 line 17-19: It can be clearly seen that the average O/C molar ratio of our laboratory-generated TB particles is very similar to that of atmospheric TBs examined by Pósfai et al. (2004), whereas it is lower than those obtained by some other authors (Tivanski et al., 2007; Chakrabarty et al., 2010; China et al., 2013).

Reviewer comment: Using microscopy technique, O and C may be also from the substrate that supports tar balls in addition to the particles themselves. Please explain how such substrate effects were considered in this study and others, because the O/C ratio is important data to compare the laboratory-generated tar balls and those from ambient.

6. Page 5 line 28-30: On the other hand, HULIS spectra contain characteristic features (a broad band at 3400–2400 cm–1 and a band of C=O at ∼1700 cm–1) suggesting the presence of carboxyl groups, whereas these bands cannot be found in the TB spectra.

Reviewer comment: This result seems to be different from that by Tivanski et al (2007), who found more carboxylic carbonyls functional groups in tar ball and concluded that tar balls are similar to HULIS. It is better to have some discussion.

7. Page 7 line 30-33: In this regard, the combination of all analytical results presents an array of supporting chemical evidence that spherical atmospheric TBs with C/O molar ratio around 10 are closer to BC in many of their properties than to weakly absorbing HULIS. In harmony with the findings of several independent studies on the

optical properties of TBs, the present results imply that TBs are indeed quite strongly light-absorbing aerosol particles and likely play an important role in the global radiation budget.

Reviewer comment: This conclusion regarding the optical properties of tar ball is too strong as the most discussion in this study focuses on their chemical properties but not on the optical properties. More discussion regarding the optical properties will be needed to conclude their optical properties. RF-IR result may be useful to this discussion (similarity of C=C?).

---

## Referee Comment (RC2) · Anonymous Referee #3 · 2 Apr 2018

Tóth et al. discuss a detailed analysis of particles generated in the lab that are supposed to mimic atmospheric tar balls. They performed several analyses including elemental analysis, Raman, FTIR, OC/EC, and pyrolysis-gas chromatography-mass spectrometry. From the results, they conclude that their TB surrogates contain a large fraction of elemental carbon, making them more similar to black carbon than to HULIS. I think the paper is nicely written and the analytical methods are sound, and it is worth publication. I have, however, a few concerns that need to be addressed before publication.

General comments:

The main issue I have with the paper is the attempt to extrapolate the findings to the properties of all atmospheric particles including the optical properties of atmospheric

tar balls. In the literature there are plenty of pieces of evidence that the properties of atmospheric tar balls are variable and therefore, the laboratory particles generated by Tóth and collaborators might be easily representing only a sub-fraction (maybe small?) of what is in the atmosphere, especially considering that there is here no discussion of measured optical properties. I will discuss more this issue in the specific comments next. I would suggest calling these "surrogates" of some TBs, not necessarily all atmospheric TBs.

Specific comments:

Abstract

Line 26: The authors should write "laboratory TBs", instead of "atmospheric TBs", because that what they measured. As mentioned earlier, the issue here is how well these laboratory-generated particles actually represent tar balls generally found in the atmosphere. More on this issue will be discussed next.

Introduction

Lines 32-34: "Since these particles. . . are able to absorb solar radiation quite efficiently in the visible (Hand et al., 2005; Alexander et al., 2008) and up to the near-IR range (Hoffer et al., 2017)". This ignores an important fraction of the literature that shows much lower absorption properties from atmospheric TBs. Neglecting to mention these works here is biasing the paper toward those studies that showed particles more similar to those discussed here. The authors should acknowledge the fact that there is a wide range in the published values of the imaginary index of refraction for atmospheric TBs and in general a large variability in the TBs properties. See for example, [1-4]. The variability in O/C ratios, for example, is well discussed in the result section on page 4, and the authors clearly acknowledge, there, that different types of TBs might exist in the atmosphere. Therefore, it is reasonable to believe that also the index of refraction values, and therefore, the absorption properties might be quite variable.

[Figure]

Lines 18-20: Similar issue here. Considering the high variability of the properties of atmospheric tar balls, it seems more logical to say here that these laboratory surrogates are similar to some of the TBs studied in the atmosphere but different from others.

Experimental section

Page 2, lines 29-31: "The concentrated aqueous phase of the tarry condensates (wood tars) was nebulised to produce tar droplets which were first exposed to a 'thermal shock' by passing them through a heated (at 650 °C) quartz tube, then cooled and dried with dry filtered air." It might be that this 'thermal shock' is resulting in TBs that represent well some atmospheric biomass burning smoke particles, but not others. A different "formation" (or transformation?) mechanism has been recently proposed for example by [3]; in that case, a thermal shock is not likely, considering that the TBs abundance increased substantially only far from the flaming region of the plume. This "delayed" formation has been shown in other studies before, as well.

Page 2, line 33: "distorted spheres" this seems in contradiction with the definition of "perfect spheres" discussed in other parts of the paper (e.g., page 1, lines 35-36). It is a bit disturbing that the not perfect sphericity is used as an argument to dismiss the study by Chakrabarty et al. in line 5 of page 2, which is one of those studies that found a week absorption for atmospheric TBs. Please be coherent.

Page 7, lines 1-2: also this high EC content points to the fact that these TBs might be at the high side of the range of absorption properties measured in the atmosphere.

Page 7, lines 2-7: How much would this artifact affect the estimated EC/TC ratio? Conclusions:

Page 7, line 32-33: Because of what mentioned earlier, I find this sentence to be biased toward those studies that found higher absorption and might not represent the large range of optical properties found in atmospheric TBs. I, therefore, suggest that the authors clearly point out this caveat to avoid providing a sense of generality that

might not be warranted.

Figure 1.: I believe that China et al. (2013) reported only the oxygen content, not the carbon. How did the authors calculate the corresponding values reported in the figure?

References

1. Chakrabarty, R.K., H. Moosmüller, L.W.A. Chen, K. Lewis, W.P. Arnott, C. Mazzoleni, M.K. Dubey, C.E. Wold, W.M. Hao, and S.M. Kreidenweis, Brown carbon in tar balls from smoldering biomass combustion. Atmospheric Chemistry and Physics, 2010. 10(13): p. 6363-6370.

2. Sedlacek, A.J., P.R. Buseck, K. Adachi, L. Kleinman, T.B. Onasch, and S.R. Springston, Tar Balls Observed in Wildfire Plumes Are Weakly Absorbing Secondary Aerosol, in ASR Science Team Meeting. 2017: Tysons, Virginia, USA.

3. Sedlacek, A.J., P.R. Buseck, K. Adachi, T.B. Onasch, S.R. Springston, and L. Kleinman, Formation and evolution of Tar Balls from Northwestern US wildfires. Atmos. Chem. Phys. Discuss., 2018. 1(28).

4. China, S., C. Mazzoleni, K. Gorkowski, A.C. Aiken, and M.K. Dubey, Morphology and mixing state of individual freshly emitted wildfire carbonaceous particles. Nature Communications, 2013. 4.

---

## Referee Comment (RC3) · Anonymous Referee #2 · 3 Apr 2018

General Comments:

Tóth et al. describes the use of multiple analytical techniques to study the organic composition of laboratory generated tar ball particles and compared them to field collected tar balls and other carbonaceous particle types including HULIS and soot particles. From these analyses, the conclude that laboratory generated tar balls are similar to some types of field tar balls based on the O:C and H:C ratios, but have much lower O:C than other possibly aged tar ball samples. Additionally, they conclude that their laboratory generated TBs are more closely related to BC than HULIS based on the O:C ratios. Overall the paper gives a good description of the laboratory generated TBs and a compelling case that they have properties between HULIS and soot. This being said, there are numerous areas that need to be addressed further in this paper to make

the comparison between their TBs and BrC/BC species. Specifically, there is some discussion about the comparison between species types, but it is lacking in some sections and the comparison data is absent in multiple figures and tables that would lead to a more polished manuscript. The IR data is of insufficient quality to make the claims in the paper and leads to questions about the quality of the Raman spectroscopy due to the low signal observed in the IR spectra for the wood tar samples. Overall the paper has a good analysis of the laboratory generated tar balls, but there seems to be some missing information that needs to be addressed.

Specific Comments:

Table 1: Please define in caption what the parenthesis represent What is the error in the individual CHNSO measurements?

Pg. 2 Line 25: "with a view to locate TBs in the light-absorbing carbon continuum" There is no analysis of the optical properties as is indicated by the last sentence of the introduction, the purpose of the paper needs to be clarified here

Pg. 2 Line 32-34: There is a description of the shape of these particles, but no actual TEM images. Please include TEM images and clarification of perfect vs. distorted spheres. An analysis of the shape factors (roundness etc.) could be used here to quantify the sphericicty

Pg. 3 Line 2: How is it calculate that 44% of the mass is collected with this stage? And what is the overall size distribution?

Pg. 4. Lines10-13: It is not stated how they believe this transformation of change in the O/C and H/C is accomplished? Could it be purely that the low volatility organics/water are driven off and what other factors could be occurring?

Pg. 4 Line35: I am not sure what "mean carbon to mass conversion factor" is telling me, perhaps showing how it is calculated would help.

Figure 1: "TB- Black locust" etc. should be labeled "laboratory generated TB" or similar

In the legend it should be noted what technique was used for analysis (e.g. EDS, CHNO, etc)

Pg. 5 Lines 7-10: The broad region around 3400cm-1 usually indicates that there is water present along with the sharp peak at 1643. Looking at Figure 2, these account for some of the peaks present in both spectra. This data indicates that there is possibly still quite a bit of water present which would possibly skew the results of the O:C analysis as well.

Figure 2: The absorption on the wood tar samples is < 0.2 a.u., which indicates a significantly lower sample loading compared to tar balls and is also a very noisy spectra below 2000 cm-1. The IR spectra needs to be improved to make any definitive statements about the carbon speciation of the wood tar and the possible presence of water needs to be addressed and corrected for.

Figure 2: It would be nice to have a comparison spectra of HULIS and soot that shows the similarities and differences since they are compared in this manuscript.

Pg. 5 line 30: Throughout the paper the "laboratory generated tar balls" becomes "tar balls" which refers to a specific natural source which this paper is showing similarities to. Clarification throughout the paper is needed as to which is being discussed.

Pg. 5. Line 34: "do not contain carboxyl groups" This is misleading based on the IR analysis, it would be better to say that they are not detected in the IR analysis

Pg. 6 Line 3: "All three types of wood tar were Raman inactive" is not substantiated because of the low noisy signal in the IR spectra demonstrating low loading of the wood tar compared to the laboratory tar balls.

Pg. 6 Lines 2-15: This needs to be more descriptive in comparison to soot as well as HULIS

Pg. 6 Line 16: Since there is already a lot of comparison between the laboratory generated tar balls and the tar starting material, why not compare at least one of these

in this section as well.

Pg. 7 Line 7: I was left wondering here how the EC/TC compared to HULIS and soot using the same method

Pg. 7 line 12: The similarity to the savanna fire data is valid for O/C only since H/C was not calculated for the savanna fires.

P7. 7 Lines 27-30: "Our results. . .combustion aerosol" this paper only shows the similarity between the laboratory generated tar balls and atmospheric tar balls, there is no data to confirm a mechanism of formation of tar balls

P7. Lines32-33: "In harmony. . .global radiation budget" please add citations to the studies on optical properties of TBs here. The main purpose of this paper was to describe the chemical composition of laboratory tar balls and the similarity to other carbonaceous particles, but there is no discussion throughout the manuscript on how they are important for light absorption (though indeed they are!).

Table 2: For the * samples (e.g. 2,4 dimethylfuran) that are only in the laboratory generated TBs, they should be excluded since it is misleading on first read through

Technical Corrections:

Pg. 2 Lines 9-16: TEM-EDS/SEM-EDS/ETEM/ESEM should all be combined into a single EDS since that is the technique used to analyze the composition

Pg. 2 Line 28: Should "chops" be "chips"?

Pg. 6 Line 7-8. "The peak fitting. . . software" should be moved to the experimental

---

## Author Comment (AC1) · 13 Jun 2018

Response to Interactive comment of Anonymous Referee #1

General comments: This study, "Chemical characterization of laboratory-generated tar ball particles", by Tóth et al., generated tar balls and characterized their chemical properties using several unique instruments. They compared tar balls with other similar carbonaceous particles (BC and HULIS) and discussed how they are different or similar. Tar ball particles are important and abundant in the atmosphere but are not well known. Thus, the characterization of tar balls is significant in this field. The manuscript is clearly written with enough data. I feel more discussion is useful regarding the similarity of the atmospheric and laboratory-generated tar balls, in addition to their morphology and O/C molar ratio. Overall, I think this manuscript includes important results on the atmospheric science.

Specific comments:

1. Page 2 Line 18-20: 'The morphological and structural characteristics and the elemental composition of the TB particles generated by this experimental system were highly similar to those of atmospheric TBs.'

Reviewer comment: Please discuss more about how they are similar to each other. Their similarity is a critical point to use the laboratory-generated tar ball data to the atmospheric tar balls. Although some such discussion may have been done in their previous studies, I think some discussion is useful here. Also, the tar ball formation process in this laboratory experiment may be different from that observed in other studies from wild fire (e.g., Posfai et al., 2004; Adachi and Buseck, 2010). They found more tar balls in aged smoke than young smoke. The laboratory-generated tar balls are emitted as primary particles and can be detected directly from the fresh smoke, suggesting that no tar ball number change as aged(?). More explanation will be useful here how they are similar to or different from ambient tar balls regarding their formation process.

Response: The sentences have been modified: The structural characteristics (homogeneity) and the elemental composition of the TB particles generated by this experimental system were highly similar to those of atmospheric TBs published by Pósfai et al. (2004) and Adachi and Buseck et al. (2011) as the Lab-TBs have homogenous internal structure without core and concentrically wrapped graphitic layers. Similarly to the atmospheric TBs the generated Lab-TB particles were also refractory under the electron beam of the TEM. Their particle diameter extended up to 360 nm measured by a DMPS system. The high C/O molar ratio (see later) of the Lab-TB particles were very similar to those found by other authors for atmospheric TBs (C/O: 8–10; Pósfai et al., 2004; Niemi et al., 2006). However, it should be noted here that other authors

(Tivanski et al., 2007; China et al., 2013) reported atmospheric TBs with significantly lower C/O ratios (1–2). Since atmospheric aging of particles is not simulated in our laboratory experiment just a fast "heat shock" is applied to the primary droplets (to which ejected particles may also be exposed in a biomass fire) therefore chemically different TBs may occur in biomass smoke plumes depending on the mechanisms and conditions of post-emission (photo)chemical transformations. Thus we believe that the observation of more tar balls in slightly aged plumes during the SAFARI experiment does not contradict the finding of our experimental study.

2. Page 2 Line 32-34: 'The shapes of the TB particles (investigated by TEM) generated from European turkey oak were mostly distorted spheres.'

Reviewer comment: Tar balls are defined as "TBs can be unambiguously identified by electron microscopy as perfectly spherical amorphous particles externally mixed in relatively fresh biomass burning plumes" in Page 1 line 35-36 or "in this paper we use the term exclusively to refer to combustion particles that share all the key characteristics that were described above" in page 2 line 5-6. However, "distorted sphere" contradicts "perfectly spherical amorphous particles." Clear definition of tar ball will be needed here, if they include those are not perfectly spherical. TEM images of European turkey oak tar ball may be useful as well as those from other sources.

Response: The sentences have been modified: page 1 line 35: 'TBs can be unambiguously identified by electron microscopy as spherical amorphous particles externally mixed in relatively fresh biomass burning plumes (Pósfai et al. 2003 and 2004, Adachi and Buseck, 2011).' The sentence (page 2 line 5) has been replaced: 'That is why in this paper we apply the term "tar balls" also to non-perfectly spherical particles.' Page 2, line 32: 'The shapes of the particles (investigated by TEM) generated from European turkey oak were mostly distorted spheres.' Page 2, line 35: The morphologies of these particles were very similar to those of the freshly formed atmospheric TBs (or to the precursor particles of the TBs) collected from biomass burning smoke 3–4 km away from fire, presented by Adachi and Buseck (2011) in their figure 4/c. Although the

shape of the particles generated from European turkey oak was not perfectly spherical on the TEM grids, we assume that prior to impaction they were likely spherical. Based on this assumption and the fact that their chemical properties e.g. elemental composition and IR spectra (see later) were similar to those of the other laboratory generated TB particles, we refer these particles also as Lab-TBs.

3. Page 3 Line 11: Reviewer comment: Please spell out "TCD."

Response: The sentence has been modified: 'The instrument was equipped with a thermal conductivity detector (TCD).'

4. Page 4 Line 10-13: 'It should be noted that wood tars (starting material for TB generation) exhibited significantly higher O/C and H/C molar ratios (0.182 and 1.215, respectively), which strongly suggests that the 'thermal shock' employed during TB generation (as described in Hoffer et al. (2016)) has markedly increased the degree of aromatisation (Francioso et al., 2011).'

Reviewer comment: Is it possible that water (H2O) is included in the wood tar to increase these molar ratios?

Response: Yes, it is also possible that some residual water is present in the raw tar which is then removed in the thermal process. Its presence may also be a plausible explanation of the markedly different shapes and C/O ratios reported for atmospheric TB particles in the literature.

5. Page 4 Line: 17-19: 'It can be clearly seen that the average O/C molar ratio of our laboratory-generated TB particles is very similar to that of atmospheric TBs examined by Pósfai et al. (2004), whereas it is lower than those obtained by some other authors (Tivanski et al., 2007; Chakrabarty et al., 2010; China et al., 2013).'

Reviewer comment: Using microscopy technique, O and C may be also from the substrate that supports tar balls in addition to the particles themselves. Please explain how such substrate effects were considered in this study and others, because the O/C

ratio is important data to compare the laboratory-generated tar balls and those from ambient.

Response: All tar balls were collected on Cu TEM grids covered by lacey Formvar+carbon films. Most particles attach to the edges of the holes in the lacey film, with most (or in some cases all) of their projected areas (and thus volumes) occurring above the holes in the film. The use of the lacey film allows us to find particles that are practically hanging in vacuum while being studied in the TEM. We took great care both in this and all former studies to analyze the compositions of only those particles that had sufficient volumes over holes, avoiding any contribution from the support film in the EDS spectra. In terms of measurement protocol, our EDS data are thus consistent over several studies spanning 15 years, starting with the ones that were published on the savanna fire emissions observed in the SAFARI experiment (Pósfai et al., 2003; and Li et al., 2003).

6. Page 5 Line 28-30: 'On the other hand, HULIS spectra contain characteristic features (a broad band at 3400–2400 cm−1 and a band of C=O at 1700 cm−1) suggesting the presence of carboxyl groups, whereas these bands cannot be found in the TB spectra.'

Reviewer comment: This result seems to be different from that by Tivanski et al (2007), who found more carboxylic carbonyls functional groups in tar ball and concluded that tar balls are similar to HULIS. It is better to have some discussion.

Response: The tar balls investiated by Tivanski et al. (2007) were similar to those investigated by Hand et al. (2005). Both authors investigated samples collected during the Yosemite Aerosol Characterisation Study (YACS, 2002) by the same sampler (time-resolved aerosol collector (TRAC)). Hand et al. (2005) described the differences between their tar balls and the tar balls reported by Pósfai et al. (2004). The tar balls from the Yosemite experiment exhibited prolonged atmospheric processing (two or more days during transport to the sampling site) than the samples collected by Pósfai et al. (2004). The difference in the atmospheric ageing can explain the differences between the two tar ball types. On the other hand tar balls investigated by Hand et al., 2005 and Tivanski et al., 2007 might have different chemical composition (lower the C/O ratio), thus the properties of particles might be more similar those of HULIS.

7. Page 7 Line 30-33: 'In this regard, the combination of all analytical results presents an array of supporting chemical evidence that spherical atmospheric TBs with C/O molar ratio around 10 are closer to BC in many of their properties than to weakly absorbing HULIS. In harmony with the findings of several independent studies on the optical properties of TBs, the present results imply that TBs are indeed quite strongly light-absorbing aerosol particles and likely play an important role in the global radiation budget.'

Reviewer comment: This conclusion regarding the optical properties of tar ball is too strong as the most discussion in this study focuses on their chemical properties but not on the optical properties. More discussion regarding the optical properties will be needed to conclude their optical properties. RF-IR result may be useful to this discussion (similarity of C=C?).

Response: We agree with the reviewer and modified the sentences accordingly: In this regard, the combination of all analytical results presents an array of supporting chemical evidence that spherical atmospheric TBs with C/O molar ratio around 10 are closer to BC in many of their properties than to weakly absorbing HULIS. Since the optical properties of the particles are closely related to their chemical composition, the finding of the present study imply (in harmony with the findings of several independent studies on the optical properties of TBs) that spherical TBs with high C/O ratio are indeed quite strongly light-absorbing aerosol particles and likely play an important role in the global radiation budget.   Response to Interactive comment of Anonymous Referee #2 Comments and questions of the reviewers are in italics Authors' answers are in regular typeface Parts of the answers highlighted in yellow are inserted into the revised manuscript.

General Comments: Tóth et al. describes the use of multiple analytical techniques to study the organic composition of laboratory generated tar ball particles and compared them to field collected tar balls and other carbonaceous particle types including HULIS and soot particles. From these analyses, the conclude that laboratory generated tar balls are similar to some types of field tar balls based on the O:C and H:C ratios, but have much lower O:C than other possibly aged tar ball samples. Additionally, they conclude that their laboratory generated TBs are more closely related to BC than HULIS based on the O:C ratios. Overall the paper gives a good description of the laboratory generated TBs and a compelling case that they have properties between HULIS and soot. This being said, there are numerous areas that need to be addressed further in this paper to make the comparison between their TBs and BrC/BC species. Specifically, there is some discussion about the comparison between species types, but it is lacking in some sections and the comparison data is absent in multiple figures and tables that would lead to a more polished manuscript. The IR data is of insufficient quality to make the claims in the paper and leads to questions about the quality of the Raman spectroscopy due to the low signal observed in the IR spectra for the wood tar samples. Overall the paper has a good analysis of the laboratory generated tar balls, but there seems to be some missing information that needs to be addressed.

Specific Comments: 1. Table 1: Reviewer comment: Please, define in caption what the parentheses represent. What is the error in the individual CHNSO measurements?

Response: The numbers in parentheses represent the relative standard deviation (RSD%) of the measurements. It was found that the error in the individual CHNSO measurements of the carbon, hydrogen, nitrogen, and oxygen content of laboratory generated TB samples was between 5–10 %, 15–20 %, 15–30 %, and 5–10 %, respectively. In case of wood tar samples the measurement error of the carbon, hydrogen, nitrogen, and oxygen was between 5–10 %, 2–5 %, 15–30 %, and 2–5 %, respectively. The caption of Table 1 has been modified: 'Oxygen to carbon (O/C) and hydrogen to carbon (H/C) molar ratios of laboratory-generated TBs and wood tar samples measured by TEM-EDS (from analysis of 12 particles from each sample) and by CHNSO elemental analyser. The numbers in parentheses represent the relative standard deviation (RSD%) of the parameters.*Data from Hoffer et al. (2017).' The text of table 1 has been complemented with: 'Lab-TBs', or 'Lab-TB samples average', and values of RSD% in parentheses as well.

2. Pg. 2 Line 25: 'with a view to locate TBs in the light-absorbing carbon continuum' Reviewer comment: There is no analysis of the optical properties as is indicated by the last sentence of the introduction, the purpose of the paper needs to be clarified here.

Response: The sentence has been modified: '...with a view to locate the chemical properties of refractory TBs (with high C/O ratio) in the continuum of carbonaceous aerosol constituents.'

3. Pg. 2 Line 32-34: Reviewer comment: There is a description of the shape of these particles, but no actual TEM images. Please include TEM images and clarification of perfect vs. distorted spheres. An analysis of the shape factors (roundness etc.) could be used here to quantify the sphericicty.

Response: TEM images of tar balls as well as the morphology of the particles produced from Norway spruce and black locust was already published and discussed in Hoffer et al. (2017). In the present study the chemical charactersitics of the very same particles were measured, since the sampling for the chemical analysis was performed during the optical measurements described in Hoffer et al. (2017). Thus the TEM images in Hoffer et al., 2017 show the morphology of the investigated particles of the present study. The TEM images of the Turkey oak, which was not published previously, show non- spherical shapes (these particles are likely less viscous and get distorted upon impaction onto the sampling grid), in this case reference is given to TEM images of similar particles already published in the literature. (see referee#1, comment#2)

4. Pg. 3 Line 2: Reviewer comment: How is it calculate that 44% of the mass is collected with this stage? And what is the overall size distribution?

[Figure]

Response: The overal number size distribution of the generated particles was measured by a DMPS system (Hoffer et al., 2017). Based on the calculated volume size distribution the relative mass of the particles on the second stage of the Berner impactor (between 125 and 250 aerodynamic diameter) was estimated. It was estimated from the DMPS data that about half of the particulate mass can be found on between 117 and 235 nm, thus also in stage 2. The sentence has been modified: 'In the case of the impactor samples we used the samples collected on stage 2 (aerodynamic diameter between 125 and 250 nm), representing about half (based on the DMPS measurements ∼37%, ∼47%, ∼59%, for the Turkey oak, black locust and Norway spruce, respectively) of the mass of the generated particles.'

One sentence is added to the experimental section: 'The size distribution of the generated particles was measured by a DMPS system (Hoffer et al., 2017).' The 'Hoffer et al., 2016' reference has been corrected to 'Hoffer et al., 2017' on page 2 line 32.

5. Pg. 4 Lines 10-13: Reviewer comment: It is not stated how they believe this transformation of change in the O/C and H/C is accomplished? Could it be purely that the low volatility organics/water are driven off and what other factors could be occurring?

Response: Of course it can not be ruled out that the volatile compounds are driven off during the thermal treatment (also see answer to question 4 of reviewer 1 above), but the appearance of the D and G band in the Raman spectra of the thermally treated samples (tar balls) indicates that aromatisation might also be a significant mechanism affecting the composition of the particles.

6. Pg. 4 Line 35: Reviewer comment: I am not sure what "mean carbon to mass conversion factor" is telling me, perhaps showing how it is calculated would help.

Response: The average carbon to mass conversion factor was calculated from the results of the elemental analysis. The particulate mass was estimated from the measured mass of the C, H, N, S and O in a sample and it was divided by the derived mass of C. This factor was and can be used to estimate the particulate mass and/or

the mass of a compound class e.g HULIS (Kiss et al., 2002) in cases when only TC measurements are available.

7. Figure 1: Reviewer comment: "TB- Black locust" etc. should be labeled "laboratory generated TB" or similar. In the legend it should be noted what technique was used for analysis (e.g. EDS, CHNO, etc).

Response: The legend in Figure 1 has been modified, the laboratory generated tar balls as well as the analytical techniques are also indicated in the legend. The reference of Santamaria et al., 2010 has been removed from the list as the authors investigated soot extracts and not soot. The sentence on page 4 line 34 has been changed to: '... close to the upper limit of those characteristic for soot (O/C: ~0.12; H/C: ~0.38)'.

The caption of the figure has been also modified: 'Van Krevelen diagram of different soot (Akhter et al., 1985; Clague et al., 1999, Collura et al., 2005), Lab-TB, wood tar and HULIS (Krivácsy et al., 2001; Kiss et al., 2002; and Salma et al., 2007) samples. The elemental compositions were measured by energy-dispersive X-ray spectroscopy (EDS), scanning transmission X-ray microscopy with near-edge X-ray absorption fine structure spectroscopy (STXM/NEXAF), or different elemental analysis techniques with or without direct oxygen measurement (EA, EA w O, EA w/o O).'

8. Pg. 5 Lines 7-10: Reviewer comment: The broad region around 3400 cm-1 usually indicates that there is water present along with the sharp peak at 1643. Looking at Figure 2, these account for some of the peaks present in both spectra. This data indicates that there is possibly still quite a bit of water present which would possibly skew the results of the O:C analysis as well.

Response: See response#4 for comments of the reviewer#1 above. For the IR measurement the samples were collected on aluminum foils, which have lower adsorption capacity of water than the quartz filter. After the sampling the samples were stored in a desiccator filled with silica gel, which might further reduce the amount of water in the samples. Of course the presence of water in the samples analysed by the IR

spectrometer cannot be ruled out, as pointed out by the reviewer.

9. Figure 2: Reviewer comment: The absorption on the wood tar samples is < 0.2 a.u., which indicates a significantly lower sample loading compared to tar balls and is also a very noisy spectra below 2000 cm-1. The IR spectra needs to be improved to make any definitive statements about the carbon speciation of the wood tar and the possible presence of water needs to be addressed and corrected for.

Response: As it was mentioned above the presence of water in the samples was minimized. The definitive statements about the carbon speciation of the wood tar have been modified in the text. The noise of the IR spectra of the wood tar samples was reduced by smoothing using the Savitzky-Golay filter.

The following sentences have been modified in the manuscript: Page 5 line 7-10: 'The FT-IR spectra of wood tars and Lab-TBs show a very broad band between 3600 and 3000 cm−1 (might be assigned to OH-stretching of phenol and/or hydroxyl groups) and a smaller band in the region between 3000 and 2780 cm−1, can be attributed to asymmetric and symmetric C–H stretching of methyl and methylene aliphatic groups (Coates, 2000; Graber and Rudich, 2006; Yang et al., 2007)'. Page 5 line 18-23: 'The possible aromatic C–C and C–H plane deformation bands in the region between 1300–1000 cm−1 overlap with the band of the C–O single bond. The broad band at 1220 cm−1 probably belongs to the C–O stretching of phenolic hydroxyl groups in FT-IR spectra of wood tar and Lab-TB samples (Coates, 2000; Yang et al., 2007), whereas the peaks at ∼920; ∼1040, ∼1110 and ∼1321 cm−1 might correspond to the C–H bending of carbohydrate; to C–O stretch in the C–OH in carbohydrate structure; to stretching of the C–O of the C–O–C linkage; and O–H bending of C–OH group, respectively (Santamaría et al., 2006; Yang et al., 2007; Cain et al., 2010; Carletti, et al., 2010; Anjos et al., 2015).'

10. Figure 2: Reviewer comment: It would be nice to have a comparison spectra of HULIS and soot that shows the similarities and differences since they are compared in

this manuscript.

Response: The IR spectra of HULIS and soot were measured in other studies which were referenced in our study (page 5, Line 25-26). The reference ('Kristensen et al., 2015') has been added to the list. The reference of 'Santamaria et al., 2006' has been corrected to Santamaría et al., 2006' on page 5: line 10; line 13; line 17; and line 22. The reference of 'Santamaría et al., 2010' has been corrected to 'Santamaria et al., 2010' on page 5 line 37 and in the reference list.

11. Pg. 5 Line 30: Reviewer comment: Throughout the paper the "laboratory generated tar balls" becomes "tar balls" which refers to a specific natural source which this paper is showing similarities to. Clarification throughout the paper is needed as to which is being discussed.

Response: All sentences have been modified to emphasize the Lab-TBs are experimented with. The modifications (e.g. 'laboratory-generated tar balls', 'tar balls' or 'Lab-TB' or 'Lab-TBs' or 'TB') highlighted in yellow are inserted into the manuscript.

12. Pg. 5 Line 34: 'do not contain carboxyl groups' Reviewer comment: This is misleading based on the IR analysis, it would be better to say that they are not detected in the IR analysis.

Response: The sentence on page 5 line 28 has been deleted: 'On the other hand, HULIS spectra contain characteristic features (a broad band at 3400–2400 cm–1 and a band of C=O at ∼1700 cm–1) suggesting the presence of carboxyl groups, whereas these bands cannot be found in the TB spectra.' The sentence (page 5 line 32) has been modified: 'Since this characteristic broad band is missing in the spectra of both wood tar and Lab-TB samples, the presence of the carboxylic groups in the samples was not confirmed.'

13. Pg. 6 Line 3: Reviewer comment: 'All three types of wood tar were Raman inactive' is not substantiated because of the low noisy signal in the IR spectra demonstrating

low loading of the wood tar compared to the laboratory tar balls.

Response: The sentences in the Abstract and on page 6 line 3 have been modified: 'Whereas Raman activity was not detected (either because of the low amount of the substances or because of their chemical composition) in the wood tar samples in the range of 1000–1800 cm−1, the Raman spectra of laboratory generated TBs were dominated by two pronounced bands with intensity maxima near 1580 and 1350 cm−1.'

14. Pg. 6 Lines 2-15: Reviewer comment: This needs to be more descriptive in comparison to soot as well as HULIS.

Response: The following sentences have been added (and modified) to the manuscript: 'Kristensen et al. (2015) investigated the Raman and IR spectra of different HULIS samples. The Raman spectra of HULIS exhibited sloping backgrounds and the presence of a small peak at 1630 cm−1 was attributed to the stretching of aromatics. The height of this peak was somewhat higher in the case of a fulvic acid standard indicating the higher aromaticity of this compound compared to the HULIS extracted from urban and rural samples. Ivleva et al. (2007) investigated the Raman spectra of a humic acid standard and those of soot samples. The obtained G and D bands were more pronounced in the spectra of these components than in the spectra of the HULIS. The Raman spectra of the macromolecular humic acid (purified, Carl Roth GmbH, Karlsruhe; Germany) investigated by Ivleva et al. (2007) was very similar to those of our Lab-TBs.'

15. Pg. 6 Line 16: Reviewer comment: Since there is already a lot of comparison between the laboratory generated tar balls and the tar starting material, why not compare at least one of these in this section as well.

Response: Py-GC-MS analysis was performed only on the laboratory generated tar ball samples and not on the wood tars, thus the comparison is not possible.

16. Pg. 7 Line 7: Reviewer comment: I was left wondering here how the EC/TC

compared to HULIS and soot using the same method.

Response: We did not analyse HULIS and soot with the same method, but according to Piazzalunga et al. (2011) the EC/TC ratio of the water soluble fraction of urban background aerosol (which contains the HULIS fraction as well) measured with the EUSAAR_2 protocol was 0.02. On the other hand Han et al. (2007) investigated the EC/TC ratio of different SRM soot and chars with the IMPOOVE TOR method. For the soot samples EC/TC ratios of 0.68–0.96 were obtained, whereas values for the char samples ranged between 0.53–0.85.

The 'Piazzalunga et al. (2011)' and 'Han et al. (2007)' references have been added to the reference list.

The name of the thermal protocol 'EUSAAR_2 long' has been changed to 'EUSAAR_2' throughout the manuscript. The 'Cavalli et al. (2010)' reference has been added to the reference list.

The caption of table 3 has been modified: 'Organic carbon (OC), elemental carbon (EC), total carbon (TC) content and ratio of elemental carbon to total carbon content (EC/TC) (RSD% of 3 Lab-TB samples) of laboratory-generated tar balls (Lab-TBs) on quartz filters (spot ø 13.06 mm) obtained by the EUSAAR_2 protocol.'

17. Pg. 7 Line 12: Reviewer comment: The similarity to the savanna fire data is valid for O/C only since H/C was not calculated for the savanna fires.

Response: The sentences have been modified: 'The elemental compositions of the Lab-TB particles generated from different wood species were very similar to one another and their O/C molar ratio was similar to those characteristic for atmospheric TBs formed in savanna fires. Page 7, line 15: The O/C and H/C molar ratios of the laboratory-generated TBs (and the O/C ratio of atmospheric TBs identified from savanna fires) are much lower than...'

18. Pg. 7 Line 27-30: Reviewer comment: 'Our results...combustion aerosol' this

paper only shows the similarity between the laboratory generated tar balls and atmospheric tar balls, there is no data to confirm a mechanism of formation of tar balls.

Response: Part of the sentence referring to the formation mechanism was deleted: 'Our results have demonstrated chemical differences between wood tars and TBs and have helped to position various types of TBs along the organic-to-graphitic carbon continuum of combustion aerosols.'

19. Pg. 7 Line 32-33: Reviewer comment: 'In harmony….global radiation budget' please add citations to the studies on optical properties of TBs here. The main purpose of this paper was to describe the chemical composition of laboratory tar balls and the similarity to other carbonaceous particles, but there is no discussion throughout the manuscript on how they are important for light absorption (though indeed they are!).

Response: The sentence has been modified (see also Referee #1 comment #7)

20. Table 2: Reviewer comment: For the * samples (e.g. 2,4 dimethylfuran), they should be excluded since it is misleading on first read through.

Response: The '2,4 dimethylfuran' has been removed from the table 2. The word 'rings' has been changed to 'ring' on page 6 line 22.

Technical Corrections: 21. Pg. 2 Lines 9-16: Reviewer comment: TEM-EDS/SEM-EDS/ETEM/ESEM should all be combined into a single EDS since that is the technique used to analyze the composition.

Response: The sentence has been modified: 'Thus, the chemical properties of TBs can only be studied by single particle analytical techniques such as TEM- or SEM-EDS (Li et al., 2003; Pósfai et al., 2003; 2004; Hand et al., 2005; Niemi et al., 2006; Adachi and Buseck, 2011; Chakrabarty et al. 2016; Adachi et al., 2017; Cong et al., 2009; 2010; China et al., 2013; Chakrabarty et al., 2006; 2010; Semeniuk et al., 2007), TEM with electron energy-loss spectroscopy (TEM-EELS) (Hand et al., 2005, Adachi and Buseck, 2011), and near-edge X-ray absorption fine-structure spectroscopy (NEXAFS)

[Figure]

using a synchrotron source (Tivanski et al., 2007).'

22. Pg. 2 Line 28: Reviewer comment: Should "chops" be "chips"?

Response: The word 'chops' has been changed to 'chips' in the manuscript.

23. Pg. 6 Line 7-8: Reviewer comment: 'The peak fitting . . . software' should be moved to the experimental.

Response: This sentence has been moved to the experimental section in the MS: 'The peak fitting of Raman spectra (in the range between 1000 and 1800 cm–1) were executed after multi-point baseline correction using by the GRAMS/AI (Version: 7.02) software.'  

Response to Interactive comment of Anonymous Referee #3 Comments and questions of the reviewers are in italics Authors' answers are in regular typeface Parts of the answers highlighted in yellow are inserted into the revised manuscript.

Tóth et al. discuss a detailed analysis of particles generated in the lab that are supposed to mimic atmospheric tar balls. They performed several analyses including elemental analysis, Raman, FTIR, OC/EC, and pyrolysis-gas chromatography-mass spectrometry. From the results, they conclude that their TB surrogates contain a large fraction of elemental carbon, making them more similar to black carbon than to HULIS. I think the paper is nicely written and the analytical methods are sound, and it is worth publication. I have, however, a few concerns that need to be addressed before publication.

General comments: Reviewer comment: The main issue I have with the paper is the attempt to extrapolate the findings to the properties of all atmospheric particles including the optical properties of atmospheric tar balls. In the literature there are plenty of pieces of evidence that the properties of atmospheric tar balls are variable and therefore, the laboratory particles generated by Tóth and collaborators might be easily representing only a sub-fraction (maybe small?) of what is in the atmosphere, especially considering that there is here no discussion of measured optical properties. I will discuss more this issue in the specific comments next. I would suggest calling these "surrogates" of some TBs, not necessarily all atmospheric TBs.

Response: We agree with the reviewer and have changed the revised manuscript accordingly. We have indicated that our experimentally generated TBs are surrogate by using the term "Lab-TB" throughout the manuscript of the revised version. We have extended the term to non-spherical particles as well, and elaborated on the possibility of the formation of various TBs having different morphologies and chemical compositions (C/O ratios) even by essentially similar mechanisms well (see our answers to comments 1 and 2 of reviewer 1). To be honest, even ourselves, using only three wood tars in a fixed experimental setup, have managed to produce TBs in two different shapes. We have referred to other studies that found TBs of markedly different morphologies and chemical (and likely optical) properties, indicating that spherical TBs of high C/O ratio are only a sub-fraction of all atmospheric TB varieties.

Specific comments: 1. Pg. 1 Line 26: Reviewer comment: The authors should write "laboratory TBs", instead of "atmospheric TBs", because that what they measured. As mentioned earlier, the issue here is how well these laboratory generated particles actually represent tar balls generally found in the atmosphere. More on this issue will be discussed next.

Response: Done as indicated above.

2. Pg. 1 Lines 32-34: Reviewer comment: "Since these particles . . .. are able to absorb solar radiation quite efficiently in the visible (Hand et al., 2005; Alexander et al., 2008) and up to the near-IR range (Hoffer et al., 2017)". This ignores an important fraction of the literature that shows much lower absorption properties from atmospheric TBs. Neglecting to mention these works here is biasing the paper toward those studies that showed particles more similar to those discussed here. The authors should acknowledge the fact that there is a wide range in the published values of the imaginary index

of refraction for atmospheric TBs and in general a large variability in the TBs properties. See for example, [1-4]. The variability in O/C ratios, for example, is well discussed in the result section on page 4, and the authors clearly acknowledge, there, that different types of TBs might exist in the atmosphere. Therefore, it is reasonable to believe that also the index of refraction values, and therefore, the absorption properties might be quite variable.

Response: We agree with the reviewer that markedly different C/O ratios imply much weaker absorption. Therefore the following sentence was added to the Introduction: It should be noted here that other authors (Chakrabarty et al., 2010; Sedlacek et al., 2017; Sedlacek et al., 2018; China et al., 2013) found less absorbing "tar ball" particles with chemical properties (C/O ratio) and optical parameters resembling those of HULIS.

The Sedlacek et al., 2017; Sedlacek et al., 2018 references have been added to the reference list.

3. Pg. 2 Lines 18-20: Reviewer comment: Similar issue here. Considering the high variability of the properties of atmospheric tar balls, it seems more logical to say here that these laboratory surrogates are similar to some of the TBs studied in the atmosphere but different from others.

Response: Done. See our response to comment#1 of Reviewer #1.

Experimental section: 4. Pg. 2 Lines 29-31: 'The concentrated aqueous phase of the tarry condensates (wood tars) was nebulised to produce tar droplets which were first exposed to a 'thermal shock' by passing them through a heated (at 650 °C) quartz tube, then cooled and dried with dry filtered air.'

Reviewer comment: It might be that this 'thermal shock' is resulting in TBs that represent well some atmospheric biomass burning smoke particles, but not others. A different "formation" (or transformation?) mechanism has been recently proposed for example by [3]; in that case, a thermal shock is not likely, considering that the TBs

abundance increased substantially only far from the flaming region of the plume. This "delayed" formation has been shown in other studies before, as well.

Response: We agree with the reviewer. The following sentence was added to the Conclusions: Since atmospheric aging of particles is not simulated in our laboratory experiment just a fast "heat shock" is applied to the primary droplets (to which ejected particles may also be exposed in a biomass fire) therefore chemically different TBs may occur in biomass smoke plumes depending on the mechanisms and conditions of post-emission (photo)chemical transformations.

5. Pg. 2 Line 33: Reviewer comment: "distorted spheres" this seems in contradiction with the definition of "perfect spheres" discussed in other parts of the paper (e.g., page 1, lines 35-36). It is a bit disturbing that the not perfect sphericity is used as an argument to dismiss the study by Chakrabarty et al. in line 5 of page 2, which is one of those studies that found a week absorption for atmospheric TBs. Please be coherent.

Response: See our response to comment#2 of referee#1.

6. Pg. 7 Lines 1-2: Reviewer comment: also this high EC content points to the fact that these TBs might be at the high side of the range of absorption properties measured in the atmosphere.

Response: Yes of course, the presence of the refractory carbonaceous compounds in those particular tar balls indicates that the optical properties of this type of tar balls are close to those of BC.

7. Pg. 7 Lines 2-7: Reviewer comment: How much would this artifact affect the estimated EC/TC ratio?

Response: As we have used the standard EUSAAR_2 protocol which is widely used for the analysis of aerosol samples, we would refrain from overruling its buil-in split-point correction in order to make an informed guess about the bias in EC/TC caused by split-point uncertainties. These uncertainties are extensively discussed in the literature. They may be partly responsible for the unexpectedly high EC readings, but the presence of highly refractory near-elemental carbon may also plays a significant role.

8. Pg. 7 Lines 32-33: Reviewer comment: Because of what mentioned earlier, I find this sentence to be biased toward those studies that found higher absorption and might not represent the large range of optical properties found in atmospheric TBs. I, therefore, suggest that the authors clearly point out this caveat to avoid providing a sense of generality that might not be warranted.

Response: The sentence has been modified: see comment #7 of Reviewer #1.

9. Figure 1. Reviewer comment: I believe that China et al. (2013) reported only the oxygen content, not the carbon. How did the authors calculate the corresponding values reported in the figure?

Response: By assuming that the particles are composed of only carbon and oxygen, their carbon content was estimated as the difference between particle mass and the mass of oxygen. The calculated C/O ratio is not affected by the presence of other elements (e.g. hydrogen) which cannot be detected by the applied EDS method.